# Characterisation of meteorological drought at sub-catchment scale in Afghanistan using station-observed climate data

**Yun Chen**（ID）*, **David Penton**（ID）, **Fazlul Karim**, **Santosh Aryal**（ID）, **Shahriar Wahid***,
**Peter Taylor**（ID）, **Susan M. Cuddy**（ID）

Environment, Commonwealth Scientific and Industrial Research Organization (CSIRO), Canberra, Australia

* Yun.Chen@csiro.au (YC); Shahriar.Wahid@csiro.au (SW)

**Data Availability Statement:** All relevant data on which our analyses based are within the manuscript and its Supporting Information. They

## Abstract

Droughts have severely affected Afghanistan over the last four decades, leading to critical food shortages where two-thirds of the country's population are in a food crisis. Long years of conflict have lowered the country's ability to deal with hazards such as drought which can rapidly escalate into disasters. Understanding the spatial and temporal distribution of droughts is needed to be able to respond effectively to disasters and plan for future occurrences. This study used Standardized Precipitation Evapotranspiration Index (SPEI) at monthly, seasonal and annual temporal scales to map the spatiotemporal change dynamics of drought characteristics (distribution, frequency, duration and severity) in Afghanistan. SPEI indices were mapped for river basins, disaggregated into 189 sub-catchments, using monthly precipitation and potential evapotranspiration derived from temperature station observations from 1980 to 2017. The results show these multi-dimensional drought characteristics vary along different years, change among sub-catchments, and differ across temporal scales. During the 38 years, the driest decade and period are 2000s and 1999–2022, respectively. The 2000–01 water year is the driest with the whole country experiencing 'severe' to 'extreme' drought, more than 53% (87 sub-catchments) suffering the worst drought in history, and about 58% (94 sub-catchments) having 'very frequent' drought (7 to 8 months) or 'extremely frequent' drought (9 to 10 months). The estimated seasonal duration and severity present significant variations across the study area and among the study period. The nation also suffers from recurring droughts with varying length and intensity in 2004, 2006, 2008 and most recently 2011. There is a trend towards increasing drought with longer duration and higher severity extending all over sub-catchments from southeast to north and central regions. These datasets and maps help to fill the knowledge gap on detailed sub-catchment scale meteorological drought characteristics in Afghanistan. The study findings improve our understanding of the influences of climate change on the drought dynamics and can guide catchment planning for reliable adaptation to and mitigation against future droughts.

can be used to reproduce the results of our study (or replicate our study's findings).

**Funding:** The research is funded by Australia's Department of Foreign Affairs and Trade and The Commonwealth Scientific and Industrial Research Organisation during 2020-21. The funders had no role in study design, data collection and analysis, decision to publish, or preparation of the manuscript. Authors received salary from the Commonwealth Scientific and Industrial Research Organisation.

## Introduction

The possibility of drought and the resultant water shortage is an ever-present hazard in arid regions like Afghanistan. Droughts have led to crises in food production and impacting on sanitation, health, resilience and livelihoods in Afghanistan. Two thirds of the country's population is in a food crisis and living in extreme poverty [1]. Understanding the spatial and temporal distribution of droughts is important to be able to plan for and respond effectively to such disasters. In Afghanistan, a severe drought event generally means low winter rainfall in two successive years. According to FAO [1], droughts have affected 6.5 million people since 2000, during four major events in 2000, 2006, 2008 and 2011. In recent years, there has been a marked tendency for this drought cycle to occur more frequently. The 2018 drought, which was one of the most severe droughts in Afghanistan in the last four decades, directly affected more than two-thirds of the country with 13.5 million people classified by the FAO as facing 'crisis' or worse levels of food insecurity and at least 0.3 million people internally displaced. Several strands of scientific analyses and experiential evidence from Afghanistan indicate the increasing frequency, scale, duration, and impacts of drought in the country; a trend that is set to intensify in the future according to all climate change scenarios for Afghanistan [1].

There are many classifications for drought, e.g. meteorological, agricultural, hydrological, and socioeconomic droughts [2, 3]. The classification depends on the temporality (e.g. permanent and seasonal drought), trigger characteristics (e.g. meteorological, hydrological and agricultural drought), statistical parameters (indices focusing on precipitation, surface and subsurface water, soil moisture, vegetation, crop yield, and such), and impact (socio-economic, ecological and environmental drought) [1]. However, meteorological drought usually instigates the other forms of drought–it can take weeks to months for precipitation (P) deficiencies to produce soil moisture decrease, streamflow reduction, drop of reservoir levels and decline in groundwater tables [4]. Thus, an initial focus on meteorological drought assessment can provide the best primary insight to drought monitoring and early warning efforts.

Meteorological drought is mainly driven by a prolonged lack of (or below average) rainfall [5], possibly intensified by hot temperature (T) causing high evapotranspiration (ET) rates [6]. There is no one definition of meteorological drought. Wilhite and Glantz [2] described it as a condition relative to some long-term average condition of balance between rainfall and evapotranspiration in a particular area. FAO [1] recast this description as the deviation from average rainfall/snowfall, calculated as the degree of dryness and the duration of the dry period because of below average precipitation. The definition of meteorological drought varies in different countries and in different periods of time and thus applying a definition which is used in one region could be inappropriate elsewhere. The importance of the meteorological drought view is that meteorological parameters, such as P and ET, could be the first indicators of drought occurrence.

It is generally recognised that drought is a regional phenomenon defined by its three characteristics: spatial extent–area and distribution, temporal variation–frequency and duration, and degree of magnitude–severity and intensity [7–11]. Drought affects many regions worldwide and future climate projections imply that drought severity and frequency will increase [1]. Hence, the impacts of drought on the environment and society will also increase considerably.

Monitoring and early warning systems for drought rely on several indicators. Drought indices (DI) have been widely used to quantify rainfall deficits, soil moisture and water availability and to assess drought severity. Many different quantitative meteorological DI have been developed and applied in drought detection and monitoring. Excellent reviews can be found in the published literature [e.g. 3, 11–14]. There is no single drought index that is ideal for all regions.

In most cases, it is useful to consider more than one index, examine the sensitivity and accuracy of the selected indices, the correlation between them, and explore how well they complement each other in the context of specific research or management objectives. An overview of the literature shows that there are two meteorological drought indices that are widely used and report good performance. The World Meteorological Organization [15] has recommended that the Standardized Precipitation Index (SPI) be used by all National Meteorological and Hydrological Services around the world to characterize meteorological droughts, e.g. [16, 17]. However, SPI is based only on precipitation and does not address the effects of high temperatures on drought conditions, such as by damaging cultivated and natural ecosystems, and increasing evapotranspiration and water stress. Its variation or extension–the Standardized Precipitation and Evapotranspiration Index (SPEI) [18]–has been developed and extensively applied. The SPEI is a standardized measure of relative local climatology and is comparable across regions with very different climates, and equally represents both wet and dry climates. It is simply constructed and can be calculated at multiple temporal resolutions [6] using both P and potential evapotranspiration (PET) deficits simultaneously. It has relatively small data requirements, while its outcomes can be easily interpreted and used in strategic planning and operational applications. Although there is not a common and straightforward definition of meteorological drought, all types of droughts originate from a lack of precipitation, and meteorological drought could refer to a reduction in precipitation that might occur along with an increase in PET. Therefore, the adoption of the SPEI to extract robust characteristics of the drought is highly recommended by many researchers [19, 20]. Most importantly, the ease of drought indices application, input data availability and geographic setting of Afghanistan were key factors in selecting drought indices for this study.

In recent decades, many studies have reported an overall global tendency toward more frequent and severe meteorological drought as derived using drought indices [21–28]. Asadi Zarch et al. [29] investigated the means to better assess drought across different aridity zones by employing both SPI and RDI (Reconnaissance Drought Index) to observed P and PET data during the last several decades (1960–2009) worldwide. Their results suggest, that, in the face of climate change, PET should not be ignored in drought modelling. Several studies also explored drought occurrence, hazard and monitoring based on drought indices. Some recent examples from the countries adjacent to Afghanistan are highly relevant. Salehnia et al. [30], Zarei et al. [31] and Qureshi et al. [32] examined the ability of SPI or RDI for assessing historical droughts at basin or provincial or national scales of Iran over decades. Al-Mamun et al. [33] identified meteorological drought prone areas in Bangladesh by using monthly precipitation data to generate SPI time series for 35 meteorological stations in the country. Zhu et al. [34] employed SPI to evaluate drought monitoring utility from satellite-based precipitation products (TRMM and GPM) over a humid river basin in China. Ahmed et al. [35] identified historical seasonal drought events (1961–2010) in a province of Pakistan based on the SPI, using global precipitation data. While very useful in demonstrating the utility of these drought indices, these studies suffer from being at coarse spatial scale (e.g. global or region), or deriving the indices based on a limited number of stations at provincial, basin or national scale, which are yet to be tested in other regions.

Afghanistan is a data-poor (or data-scarce) country, in part because of the various wars it has endured over the last 40 years. Despite decades of experiencing and coping with drought, data and analysis regarding metrological drought impact in Afghanistan continue to be scarce [1]. In Afghanistan, published studies include drought characterization for snow-dominated regions from 1957 to 2002 [36] and for irrigated agriculture area from 1979 to 2015 [37], as well as a national-scale drought assessment during cropping seasons for the period 1901–2010 [38]. More recently, Saha et al. [39] implemented the South Asia Drought Monitor (SADM) to

investigate agricultural drought from 1979 to 2018 for four countries including Afghanistan. The system contains a set of drought indices (including SPI and SPEI) to produce seasonal outlook on potential drought hazards. However, some of the above-mentioned pioneer studies used out-of-date long-term historical data; some used gridded global datasets at a coarse resolution of 0.5˚ (or ~ 50km) [20, 36, 38]; and some were conducted only at river basin scale [36, 37]. A more detailed review of relevant applications in Afghanistan and its adjacent countries can be found from Qutbudin et al. [38]. More investigations on national scale meteorological drought specifically calibrated to Afghanistan using ground observations are needed, at a scale commensurate with the availability of the ground observation data. For this purpose, we proposed to adopt a sub-catchment conceptualisation as would be commonly used by the catchment modelling community. Using sub-catchments makes it possible to link to sub-catchment state and processes, and not be reliant on aggregating blocks (grids) or interpolating from basin-scale [40]. To the best of our knowledge, a national meteorological drought assessment based on derivation of meteorological drought characteristics at a spatial scale of sub-catchment does not exist for Afghanistan.

Therefore, the objective of this study was to assess the spatio–temporal variations and trends of meteorological drought at sub-catchment scale across various aridity zones in Afghanistan using the popular SPEI drought index. To achieve this, the multiple aspects of extent of drought occurrence or frequency, duration and severity were estimated and mapped as essential characteristics with the focus on available observed station data sources during the period of 1980 to 2017. Over four decades of conflict in the region, combined with drought and environmental mismanagement, have resulted in widespread degradation of Afghanistan's natural resources. In addition to addressing the spatial scale gap in drought assessment in Afghanistan, we hope that our study will benefit the nation's future development, especially the management of its water resources, through providing a sound basis for quantifying drought hazard and mapping drought risk at a more detailed level (temporal and spatial scales) across the whole country.

## Study area and data

### Description of study area

Afghanistan is located in southern-central Asia (Fig 1). The country and its surrounds are considered the 'crossroads of Asia'. Occupying 652,000 km$^2$, Afghanistan is a mountainous country with plains in the north and southwest. It is mostly rugged with some unusual mountain ridges accompanied by plateaus and river basins. Afghanistan is mostly outside the monsoon zone and has four seasons: spring, summer, autumn and winter though these vary greatly between topographic regions. The largest part of the country has a dry continental climate characterised by harsh winters with temperatures ranging between −15˚C to −26˚C in January, and hot summers with temperatures ranging between 35˚C and 43˚C in July. The country is generally arid in the summers, with most rainfall falling between December and April. The wet season starts in October and lasts up to June. The lower areas of northern, western and southwestern Afghanistan are the driest, with precipitation more common in the east [41]. Over the last several decades, drought conditions have appeared with more frequency. Since 1970, drought has been recorded in 1970–71, 1998–2006 (with 2000–02 being the most severe), and more recently 2017–18.

Despite having numerous rivers and reservoirs, large parts of the country are dry. There are five major river basins, Helmand, Harirod-Murghab, Kabul, Northern and Panj Amu, with areas varying from 71,995 km$^2$ (Northern Basin) to 327,660 km$^2$ (Helmand Basin). They can be further divided into 41 catchments consisting of 189 sub-catchments with an average area

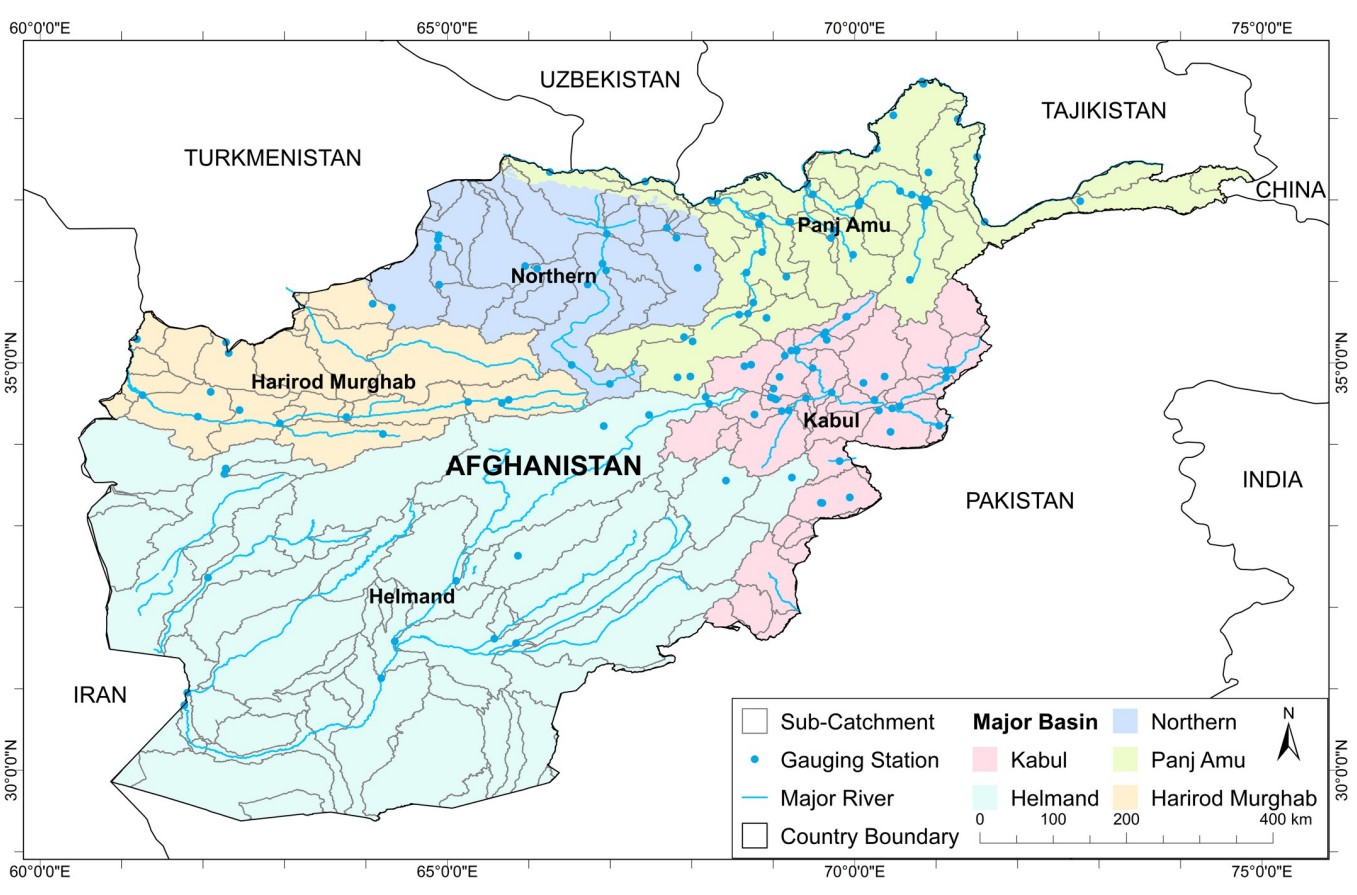

**Fig 1. Location map of Afghanistan.**

of 3,430 km$^2$ (Fig 1) within Afghanistan's borders. The endorheic Sistan catchment in Helmand Basin is one of the driest regions in the world. Kabul River which flows in an easterly direction to the Indus is the longest river in the country (560 km). With its water year starting in October, Afghanistan's snow season is between October and April, depending on altitude. Pamir Mountains and its westernmost extensions receive heavy snowfall during the winter, and the extensive snow melt in the spring season enters the rivers, lakes, and streams. However, two-thirds of the country's water flows into the neighbouring countries. Irrigated agriculture is largely dependent on having enough snow in the mountains to melt in the spring. In many places, irrigated agriculture is totally dependent on available snow in the mountains. In the lowlands rain falls mainly in the autumn and at the beginning of the year. The spring rains are of great importance to agriculture. Sufficient rain at the right time of year is a prerequisite for the rain-fed agriculture system that dominates northern Afghanistan.

## Input data

**Catchment and sub-catchment boundaries.** We delineated the five river basins of Afghanistan into 189 sub-catchments, more than 120 of them having an area of less than 2,500 km$^2$ (smaller than a 50x50km grid cell). The boundaries were generated based on 3-arc second (approximately 90 m) SRTM (Shuttle Radar Topography Mission) using ArcGIS Hydro Tools and river network and then refined based on the locations of streamflow gauges and dams (Fig 1).

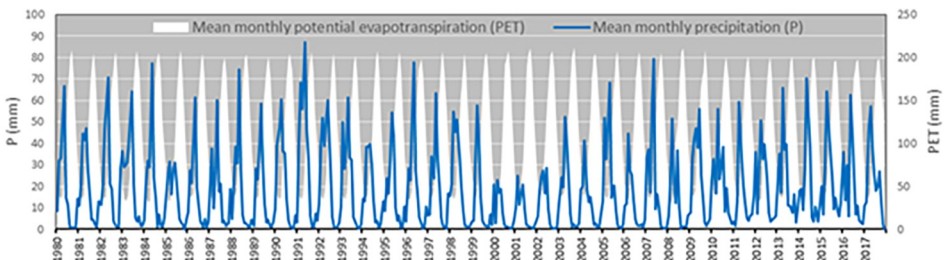

**Fig 2.** Mean monthly P (left axis) and PET (right axis) time series at national scale over the study period 1980–2017.

**Monthly precipitation (P).** Guttman [42, 43] noted that if a sufficiently long time series is presented, the results of the probability distribution will be more robust because more samples of extreme wet and extreme dry events are included. Ideally the time series should have a minimum of 30 years of data even when missing data are accounted for. Therefore, this study used 38 years (1980–2017) of monthly observed rainfall data. Daily observations for 157 stations were available from the AQUARIUS Time-Series [44] system used by the national water authority for managing its time-series data. For filling data gaps (as not all sub-catchments contain an observation station), the IDW (inverse distance weight) algorithm was used to interpolate based on area covered by the nearest station using Thiessen Polygons (also known as Voronoi diagrams). Daily data were then summed to end-of-month totals for each sub-catchment. The spatio-temporal distribution of input P data are summarised in Figs 2 and 3. In Fig 2, the monthly P and PET time series are averaged across all sub-catchments to provide P and PET at national scale.

**Monthly temperature.** Monthly minimum and maximum temperature data were obtained from the AQUARIUS Time-Series system. These observations were divided into two classes: those in the period prior to automated telemetry and those in the period with automated telemetry (in the 2000s). For the period with automated telemetry, only the automated readings were used and the sub-daily values were aggregated to daily max/min values. For the period without automated telemetry, only data for those stations that were later automated were used. A secondary comparison of the early and later period was conducted. Whenever there was a big difference (e.g. 3–5˚C warmer in the later period) between the early period and the late period, the early data were excluded. The Random Forest/Inverse Distance Weight

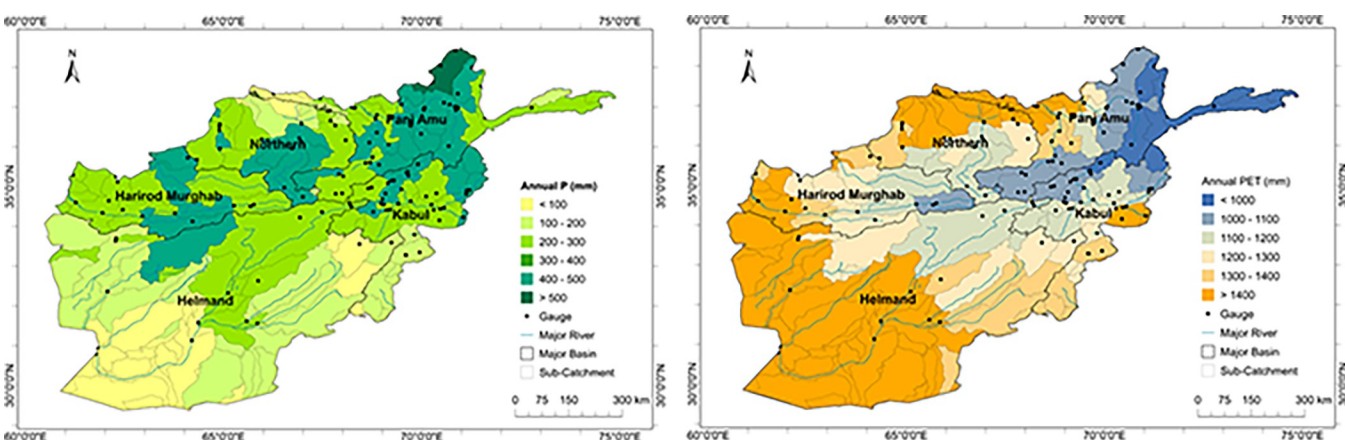

**Fig 3.** Mean annual P (a) and PET (b) at sub-catchment scale over the period 1980–2017.

**Table 1. Results for minimum and maximum temperature from the tenfold-cross validation.** The temperatures highlighted (in bold) identify the best results.

| Temperature | Validation | 1 | 2 | 3 | 4 | 5 | 6 | 7 | 8 | 9 | 10 |
|---|---|---|---|---|---|---|---|---|---|---|---|
| Minimum | $R^2$ | 0.7226 | 0.8039 | 0.8663 | 0.8802 | **0.9389** | 0.9203 | 0.9208 | 0.8801 | 0.3547 | 0.9302 |
| Maximum | $R^2$ | 0.9472 | 0.9288 | **0.9490** | 0.9168 | 0.9046 | 0.9446 | 0.9174 | 0.9121 | 0.8988 | 0.9120 |

(RF-IDW) method was applied to spatially interpolate the station data using station observations, latitude, longitude and elevation in a similar vein as Tan et al. [45]. In total, there were 73 stations trained over 14,548 days (1979-01-01 to 2018-04-03). Tenfold cross validation was performed during the training of the model. Results from the cross-validation are shown in Table 1 with best results highlighted in bold.

## Methodology

### Potential evapotranspiration (PET) calculation

Monthly PET for each sub-catchment was derived from observed minimum and maximum temperature data using the temperature-based Hargreaves and Samani (HS) empirical method [46] (Eq 1):

$$\text{ET}_\text{o} = 0.0135 k_{R_s} \frac{R_a}{\lambda} \sqrt{T_{max} - T_{min}}(T + 17.8) \tag{1}$$

where $R_a$ is the extra-terrestrial radiation (MJ m$^{-2}$ day$^{-1}$), computed for any given day as a function of the latitude of the site [47] and $\lambda$ is the latent heat of vaporization (MJ kg$^{-1}$) for the mean air temperature, $T$ (°C), that is commonly assumed equal to 2.45 MJ kg$^{-1}$. 0.0135 is a factor for conversion from American to the International system of units and $k_{R_s}$ is the empirical radiation adjustment coefficient (°C$^{-0.5}$). In the common version of HS equation, the value $k_{Rs} \approx 0.17$ is used [48]. Spatio-temporal distribution of input PET data is summarised in Figs 2 and 3. An improvement on the calculation of PET would be to use Penman-Monteith, which is always recommended for SPEI calculation. There can be significant differences amongst the SPEI series estimation by the different PET methods such as larger values in semi-arid to mesic regions and smaller values in humid regions. Unfortunately, station data were not available to apply Penman-Monteith in this study.

### SPEI derivation

SPEI is designed to consider both precipitation (P) and potential evapotranspiration (PET) in determining drought. The use of PET by SPEI gives a better representation of the full water balance of the region and provides an improved indication of the drought severity when compared to SPI. Thus, SPEI captures the main impact of increased temperatures on water demand with high sensitivity and resilience. SPEI can be calculated for an accumulation period of interest (e.g., 1, 3, 6 and 12 months) using monthly precipitation (P) and potential evapotranspiration (PET) time series (1980–2017) for each sub-catchment.

The computation procedure of SPEI follows the method outlined in [18]. The two main steps are (1) to determine the accumulation of water deficit $D = (P–PET)$ at different timescales, and (2) to normalize $D$ into a log-logistic probability distribution to obtain SPEI using Eqs 2 and 3:

$$F(x) = \left[ 1 + \left( \frac{\alpha}{x - \gamma} \right)^{\beta} \right]^{-1} \tag{2}$$

**Table 2. Drought index and characteristics analysed in this study.**

| Index/Characteristics | Definition/Description | Classification | |
|---|---|---|---|
| SPEI | Standardized Precipitation and Evapotranspiration Index | Extreme Drought (ED) | SPEI: $\leq$ –2.0 |
| | | Severe Drought (SD) | –2.0 to –1.5 |
| | | Moderate Drought (MOD) | –1.5 to –1.0 |
| | | Mild Drought (MID) | –1.0 to –0.5 |
| | | Near Normal (NN) | –0.5 to 0.5 |
| Drought occurrence (Unit: number of months) | Number of months with/of MOD or SD or ED | | SPEI: $\leq$ –1.0 |
| Drought frequency (Unit: month) | Total number of drought months under each specified timescale | Sometimes | Month:1 to 2 |
| | | Often | 3 to 4 |
| | | Frequent | 5 to 6 |
| | | Very frequent | 7 to 8 |
| | | Extremely frequent | 9 to10 |
| Drought duration (Unit: month) | Max. number of consecutive drought months during each year | Short | Month:1 |
| | | Medium | 2 to 3 |
| | | Long | 4 to 6 |
| | | Very long | 7 to 9 |
| | | Extremely long | 10 to 12 |
| Drought severity | Cumulative sum of all the SPEI values over the drought duration | Extreme | SPEI: –25 to –20 |
| | | Very high | –20 to –15 |
| | | High | –15 to –10 |
| | | Moderate | –10 to –5 |
| | | Low | –5 to –1 |

$$\text{SPEI} = \text{W} - \frac{c_0 + c_1 W + c_2 tW^2}{1 + d_1 W + d_2 W^2 + d_3 xW^3} \tag{3}$$

where $\alpha$, $\beta$ and $\gamma$ are scale, shape, and origin of the Pearson Type III distribution, respectively, of $F(x)$ which is the probability distribution function of the $D$ series. $c_0$, $c_1$, $c_2$, $d_1$, $d_2$ and $d_3$ are constants having values of 2.515517, 0.802853, 0.010328, 1.432788, 0.189269 and 0.001308, respectively. For $P(D) \leq 0.5$, $\text{W} = \sqrt{-2\ln(P)}$, and $P(D) = 1-F(x)$; for $P(D) > 0.5$, $P(D)$ is replaced by $1-P(D)$ and the sign of SPEI is reversed.

The average value of SPEI is 0, and the standard deviation is 1. The SPEI is a standardized variable, and it can therefore be compared with other SPEI values over time and space. An SPEI of 0 indicates a value corresponding to 50% of the cumulative probability of $D$, according to a log-logistic distribution. The resulting numeric values of SPEI represent the number of standard deviations that cumulative $D$ deviates from the long-term average, that is, a monthly SPEI equal to –2.0 in May 2000 means that the accumulated $D$ in that month is 2 standard deviations smaller than the long-term average, 1980–2017 in this study, of monthly $D$ recorded in May.

The categorization of drought classified by the SPEI is show in Table 2 where drought was identically divided into five levels [49]: near normal (NN), mild drought (MID), moderate drought (MOD), severe drought (SD), and extreme drought (ED). In our study, a threshold of –1.0 is used to identify (moderate, severe, and extreme) drought events/months using the SPEI index for any given sub-catchment; that is, increasingly severe meteorological droughts are indicated as SPEI decreases below –1.0. Because SPEI values are in units of standard deviation from the long-term mean, they can be used to compare meteorological anomalies for any geographic location and for any number of timescales. In general, the SPEI values at the 1-month

time scale can reflect the slightest changes in drought; at the 3-month time scale, they can represent seasonal drought; and at the 12-month time scale, these indices can imply the interannual variation in drought and long-term drought monitoring results. Therefore, this study estimates and analyses the SPEI values at 1-, 3- and 12-month time scales for each sub-catchment to indicate the monthly, seasonal and annual drought conditions in Afghanistan. Note that the name of them is modified to include the accumulation period. Thus, SPEI-1, SPEI-3 and SPEI-12 refer to accumulation periods of one month, three months and twelve months for SPEI, respectively.

## Drought characterisation

Generally speaking, SPEI-1 computed for shorter accumulation periods (1 month) can be used as an indicator for immediate impacts such as reduced soil moisture, snowpack, and flow in smaller creeks. SPEI-3 computed for medium accumulation periods (3 months) can be used as an indicator for reduced streamflow and reservoir storage. SPEI-12 computed for longer accumulation periods (12 months) can be used as an indicator for reduced reservoir storage and groundwater recharge. This study characterised drought at sub-catchment scale through estimating (1) long-term average drought occurrences (as a proxy of interannual variation and long-term change) and mean annual drought frequency (as a proxy of flash and short-term drought) based on SPEI-1 and SPEI-12 across 38 years (1980–2017), and (2) annual drought

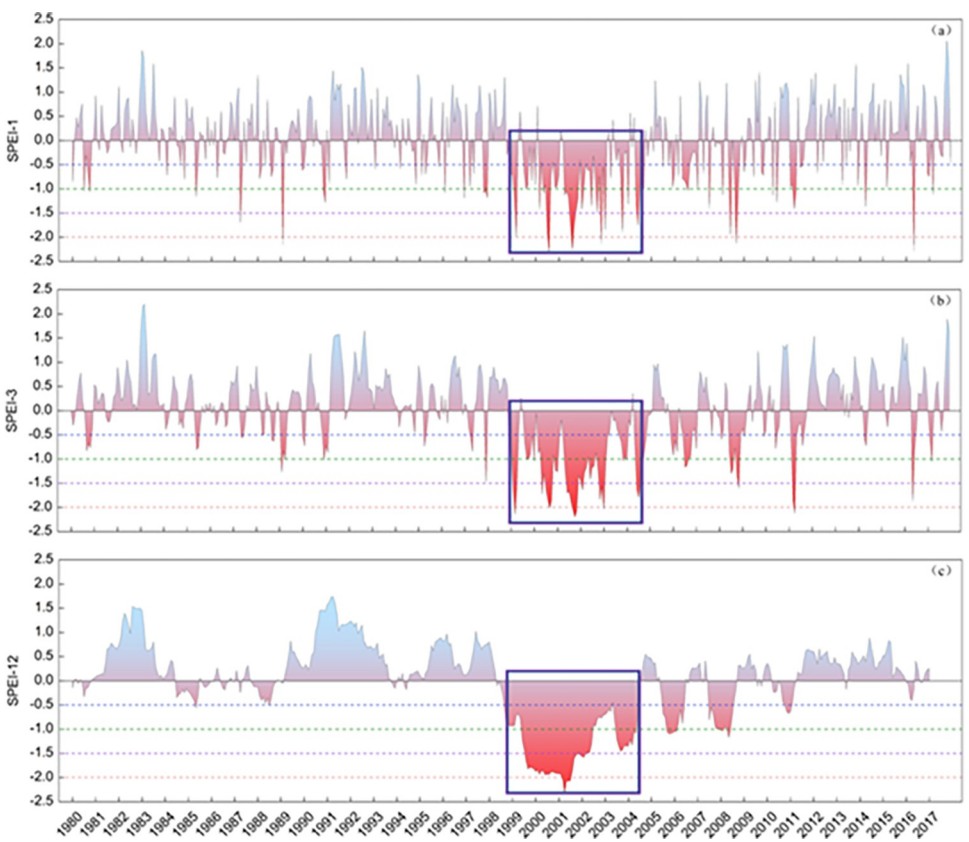

**Fig 4.** Temporal distribution of the mean SPEI at sub-catchment scale in Afghanistan for each year from 1980 to 2017: (a) monthly (SPEI-1), (b) seasonal (SPEI-3) and (c) annual (SPEI-12). The prolonged drought period from 1999 to 2004 is highlighted. Blue dotted line = Mild Drought, green dotted line = Moderate Drought, purple dotted line = Severe Drought; and red dotted line = Extreme Drought.

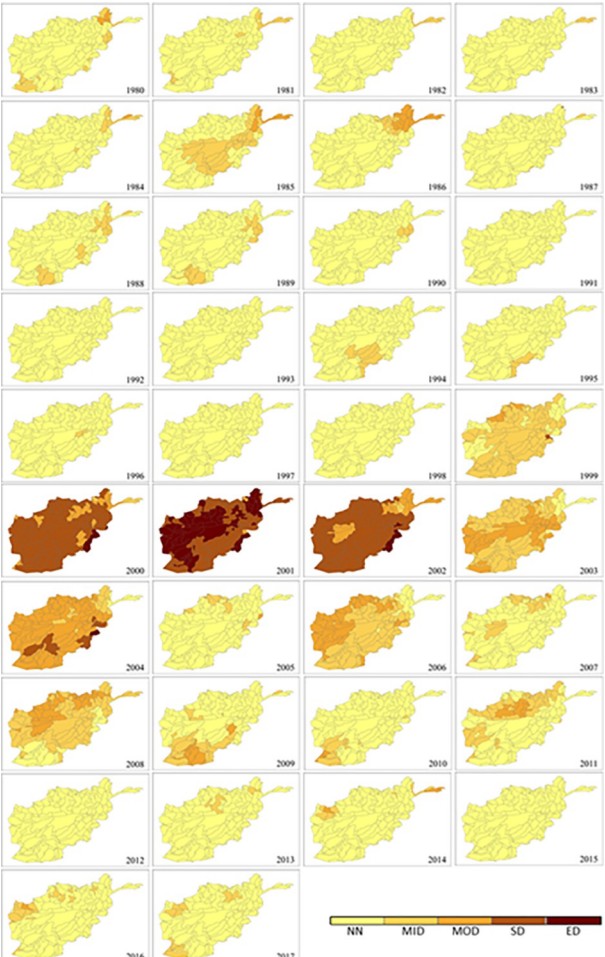

**Fig 5. Spatial distribution of the mean annual drought index (SPEI-12) at sub-catchment scale in Afghanistan for each year between 1980 and 2017.**

duration and associated severity (as an indicator of intensity) from both SPEI-1 (as a proxy of monthly drought) and SPEI-3 (as a proxy of seasonal drought). Drought intensity has a similar meaning to drought severity; as such only severity was assessed. Details of the drought characteristics that were estimated using SPEI to quantitatively describe the spatiotemporal variation of drought at sub-catchment scale in Afghanistan are listed in Table 2.

## Results and discussion

### Results

**Drought occurrence and frequency.** The broad overview of temporal distribution and spatial extent of drought occurrence at monthly, seasonal and yearly scale is presented in Fig 4. During the period 1980–2017, the results of SPEI-1, SPEI-3 and SPEI-12 (Fig 4) show that the 2000s are the driest 10 years among the nearly four decades in this study with the most prolonged drought period running from 1999 to 2004 (highlighted in the frames). The spatial distribution of SPEI-12 (Fig 5) also clearly illustrate that, in terms of percentage area of the total country, 2001 is the driest year with the whole country experiencing 'severe' to 'extreme' drought and more than 53% (87 sub-catchments) suffering the worst in history, followed by

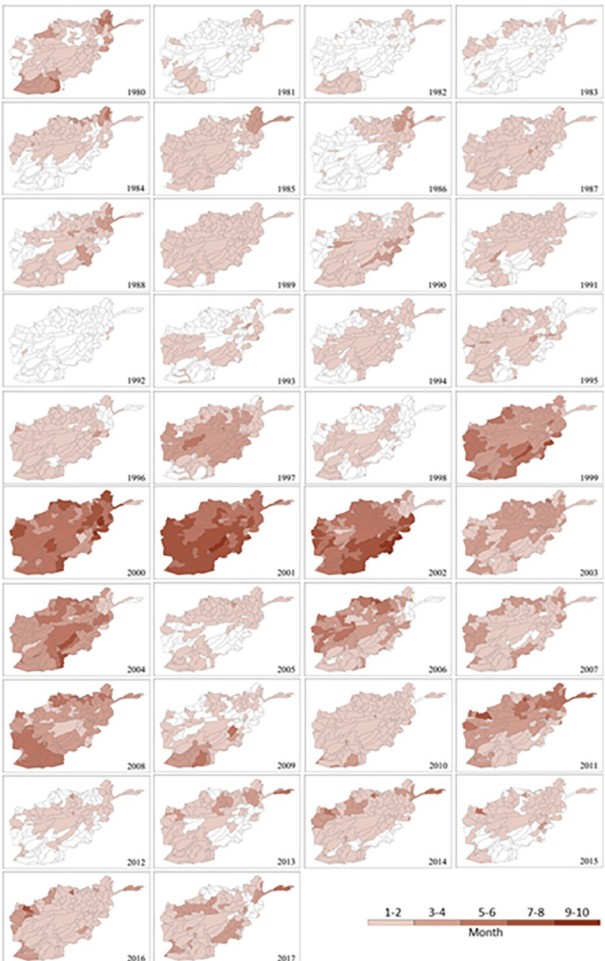

**Fig 6. Spatial distribution of mean annual drought frequency at sub-catchment scale in Afghanistan for each year between 1980 and 2017 based on SPEI-1 (monthly scale).**

'severe' to 'extreme' drought in 2000 (87%) and 2002 (86%), 'moderate' to 'severe' drought in 2004 (90%), as well as some 'moderate' drought in 2003 (37%), 2006 (33%) and 2008 (22%).

Similar to the patterns in Fig 5, the spatial distribution of annual drought frequency in Fig 6 reveals that the 2000s was the decade with the most 'frequent' drought months (5 to 6 months) per year. The peak period was 1999–2002 followed by 2004, 2006, 2008 and 2011. 2000 was the driest year in the monsoon season with increasingly 'very frequent' to 'extremely frequent' drought across much of the country (82% area, 155 sub-catchments). In the driest year of 2001, almost sub-catchments (174) across the country had at least 'frequent' drought. Of the 189 sub-catchments, 94 had 'very frequent' drought (7 to 8 months) or 'extremely frequent' drought (9 to 10 months), which covers 58% of Afghanistan, mainly located in the south of the country. This dryness prevailed until 2002 when 76% of the country (145 of 189 sub-catchments) was still classified as 'frequent' drought (46% area), and 'very frequent' to 'extremely frequent' drought (30% area).

**Drought duration.** The maximum duration of drought events identified with SPEI-1 and SPEI-3 for the 1980–2017 period is mapped in Figs 7 and 8, respectively. They reflect monthly and seasonal soil moisture conditions and provide an estimation of full water balance indicated by both P and PET. The longest ('very long' or 'extreme') durations of drought events

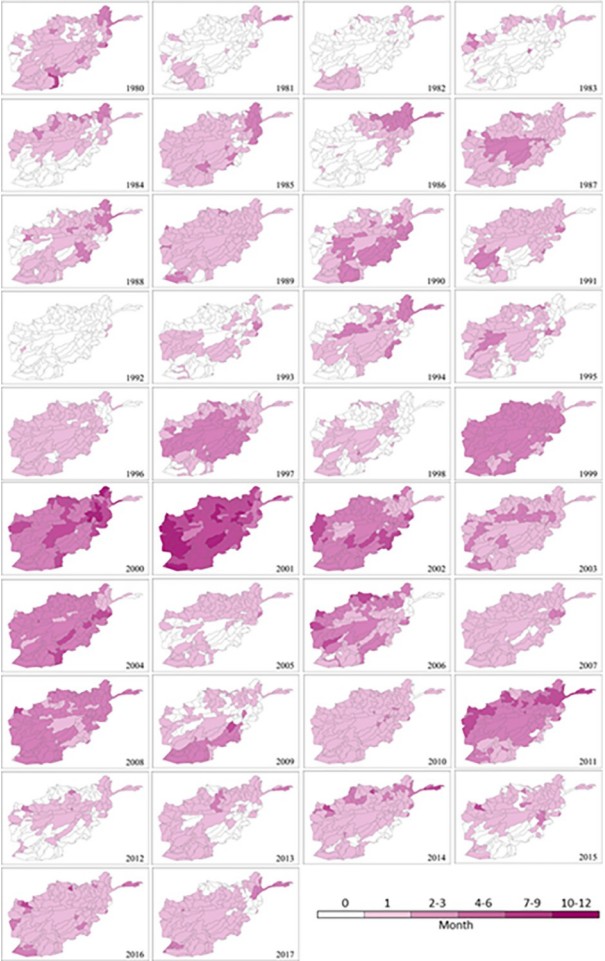

**Fig 7. Spatial distribution of mean annual maximum drought duration at sub-catchment scale in Afghanistan for each year between 1980 and 2017 based on SPEI-1 (monthly scale).**

are between 1999 and 2002, followed by 2004, 2006, 2008 and 2011. The distribution of drought duration in Fig 7 corresponds closely to that of drought frequency in Fig 6 due to their common SPEI-1 basis and close correlations. There is a similar overall spatiotemporal distribution pattern between monthly (SPEI-1; Fig 7) and seasonal (SPEI-3; Fig 8) maps. Nearly all sub-catchments show an increment in drought duration from SPEI-1 to SPEI-3 timescale during the above-mentioned years (Fig 9A). However, there are differences in their actual duration. The results of SPEI-3 show longer drought durations compared to SPEI-1 with most of the country in 'very long' seasonal drought in 2001 and 'extremely long' seasonal drought in 19% of Afghanistan (53 sub-catchments) in the southeast regions until 2002.

## Drought severity

The maximum severity of monthly (SPEI-1) and seasonal (SPEI-3) drought durations for the 1980–2017 period is mapped in Figs 10 and 11, respectively. Severity measures both the duration and degree of drought. In general, both maps, in particular Fig 11, present spatial patterns of drought severity similar to corresponding timescales of duration maps in Fig 9. The most severe ('extreme') drought conditions are observed in the 1999 to 2004 period for SPEI-3 based seasonal durations,

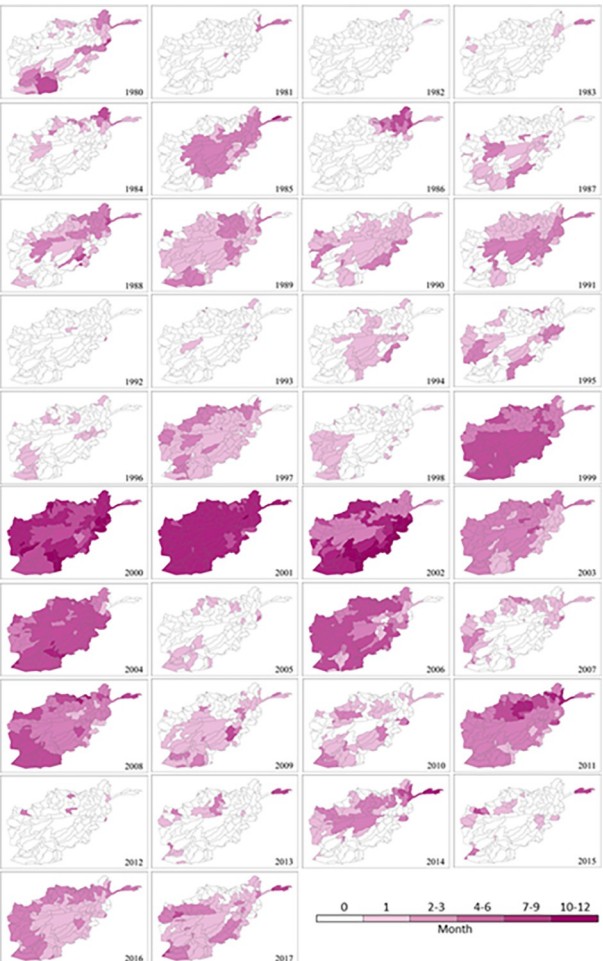

**Fig 8. Spatial distribution of annual maximum drought duration at sub-catchment scale in Afghanistan for each year between 1980 and 2017 based on SPEI-3 (seasonal scale).**

following by 2006, 2008 and 2011. However, in Fig 10 the distributions of drought severity are different from relevant timescales of duration maps in Fig 7. The most severe ('extreme') droughts are concentrated in the 2000–02 period for SPEI-1 derived maximum durations although 'moderate' severity droughts are also recorded for a small number of sub-catchments in 1997, 2004, 2006, 2007, 2008 and 2011. In agreement with the overall distribution of drought duration, a more even pattern of severity exists in the monthly SPEI-1 maps (Fig 10) compared to the seasonal SPEI-3 maps (Fig 11). However, Fig 11 shows seasonal droughts as being much more severe ('very high' or 'extreme') than the short-term (monthly/flash) drought depicted in Fig 10. The most severe monthly ('high' and 'very high') and seasonal ('very high' and 'extreme') drought severity are observed across the country in 2001 (Figs 10 and 11, respectively). The second worst year is 2000 which is significantly affected by severe droughts which is also shown in Fig 9B.

## Discussion

Afghanistan, being in a semi-arid to arid zone, is prone to droughts with prolonged droughts commonly occurring. Sub-catchments in the nation's southeast region are already chronically drought impacted. As a proportion of Afghanistan's total land surface, the area affected has expanded considerably as a result of climate change during the last two decades [1].

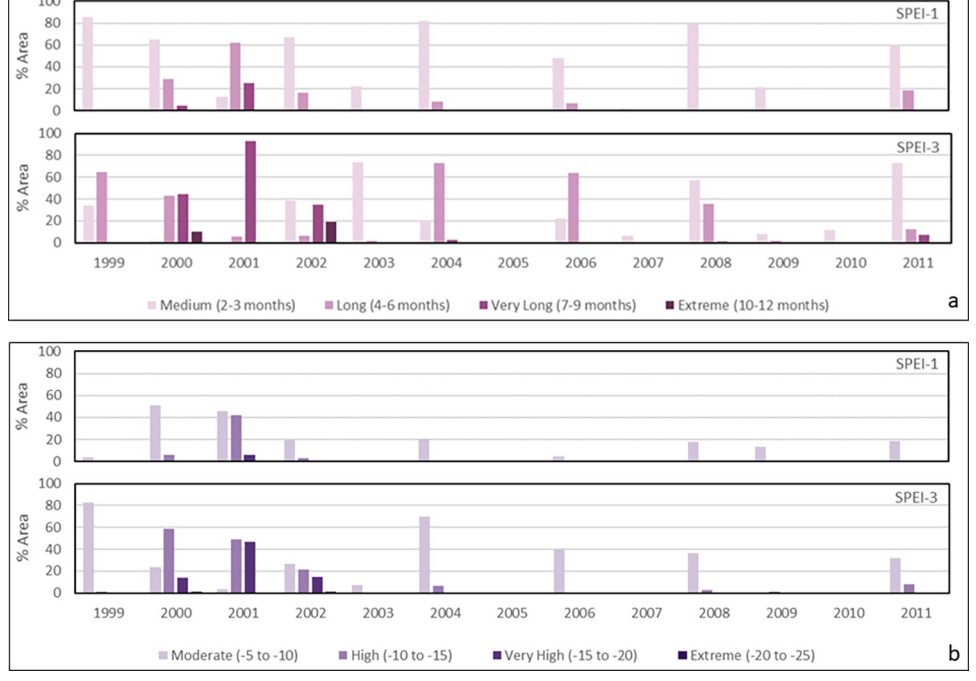

**Fig 9.** Areas (1999–2011) associated with the categories of (a) medium-to-extremely long drought duration, and (b) moderate-to-extreme drought severity.

Meteorological drought has also been increasing as a consequence of reduced spring precipitation and increased evaporation rates across the sub-catchments in the north and central regions. The increase in frequency, duration and severity of drought events during the extensive drought periods between 1999 and 2002 are mostly driven by a significant decrease in precipitation in addition to the highest PET. Progressive reduction of rainfall could have accelerated the other droughts between 2004 and 2011. This is in agreement with previous studies in Afghanistan [36, 37, 50, 51]. The southern parts of the country have been strongly affected by climatic changes, especially in winter; and the northwest and the southwest of the country have been more affected by droughts due to alternations in rainfall and temperature [38]. The droughts characterised in the present study are also consistent with the findings of other studies conducted in nearby countries and regions such as India [52], South Asia [39], central Southwest Asia [53] and Asian Least Developed Countries [54]. The onset of the 2001 drought was in 1999 with a significant decrease in rainfall, reached its peak in 2001 due to the combination of extremely low rainfall and high temperature (or PET), and persisted until 2002 when heavy snow began falling. This resembles patterns in its neighbouring countries including Pakistan [55, 56] and Iran [57, 58]. As shown in Fig 8, the most recent drought in 2011, especially severe in northern and eastern Afghanistan, affected 182 sub-catchments accounting for 93% of the area of the country. Drought is known to have direct impacts on livelihoods and the economy and has already pushed 12 million people in 14 of the country's 34 provinces into food insecurity and poverty [59].

There are certain limitations of these results that need to be taken into consideration. This study is not temporally up-to-date due to unavailability of observation data. The spatial density of monitoring stations varies across the country (Fig 1). As a result, there will be regions where there is higher uncertainty in the interpolated results. This is perhaps most noticeable in the Helmand basin–the largest basin also with the least number of stations. In addition, these results

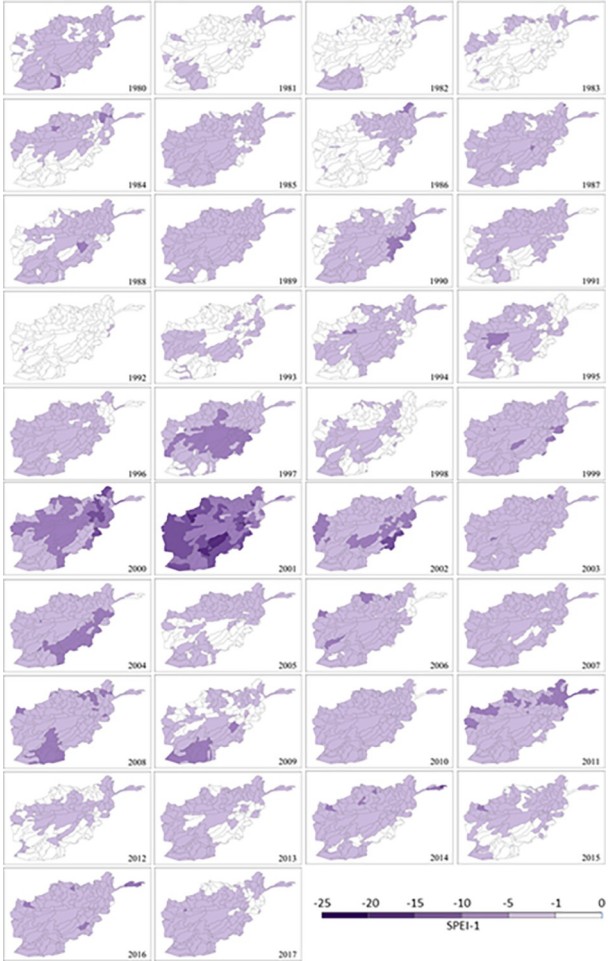

**Fig 10. Spatial distribution of annual maximum drought severity at sub-catchment scale in Afghanistan for each year between 1980 and 2017 based on SPEI-1 (monthly scale).**

do not consider snowfall and snowmelt, with the focus on meteorological drought. A hydrological drought indicator would include aspects of snowmelt but were not in scope for this study.

## Conclusions

This study characterises meteorological drought in Afghanistan at sub-catchment scale using a drought index derived from observed P and PET data. It maps the spatiotemporal extent and distribution of drought occurrence, frequency, duration and severity between 1980 and 2017 based on monthly, seasonal and yearly SPEI. The main findings are:

1. The SPEI can be used for mapping spatial distribution, temporal variation, and intensity estimation. These multi-dimensional drought characteristics vary along different years, change among sub-catchments, and depend on the SPEI timescale used for carrying out the analysis.

2. The 2000s is the decade with the most significant increase in drought extremes during the study period of 1980–2017. Drought frequency, duration and severity have increased since 1999 onwards. The driest period is between 1999 and 2002 and the driest year is 2001.

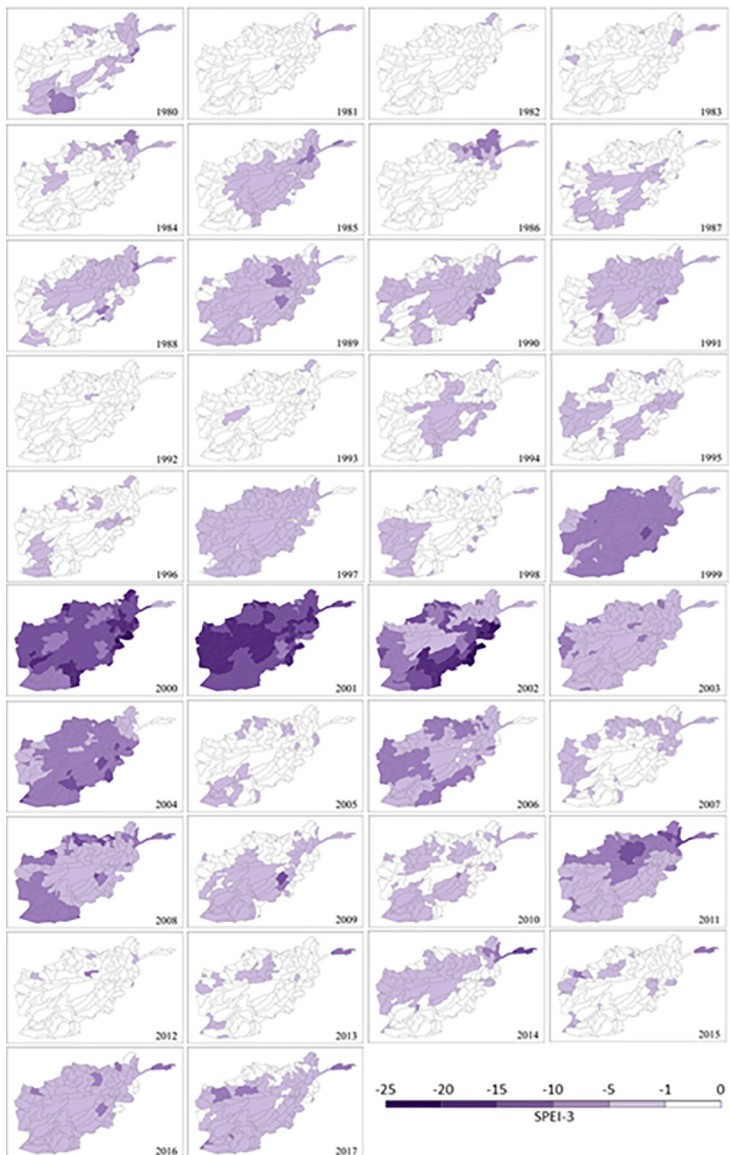

**Fig 11. Long-term annual maximum drought duration at sub-catchment scale in Afghanistan (1980–2017) based on SPEI-3.**

Other severe droughts are identified in 2004, 2006, 2008 and 2011. Seasonal drought has a much more significant change pattern than monthly drought across the 38 years.

3. Afghanistan experienced the worst drought in its recorded history in 2001 with frequency, duration and severity reaching their extremes across almost all sub-catchments in Afghanistan. Its onset was in 1999 and it prevailed until 2022, affecting most of the country. The lowest annual precipitation during 1999–2000 has led to the reduction of water and soil moisture, and consequently aggravated the impact of the peak drought in 2001.

4. The nation suffers from recurring droughts with varying length and intensity. There is a trend toward increasing drought condition with extended duration and higher severity from southeast to north and central of the country over all sub-catchments. The most severe

drought in recent decades witnessed in 2011 has hit harder in the northern sub-catchments than in the south.

5. Station-based P and PET are the most valuable data for drought monitoring in developing countries such as Afghanistan. However, there may be gaps and errors induced by human mistakes or uncontrolled interruptions. Extra caution, specific assessment and proper techniques for gap-filling and error correction are recommended before applying these datasets for drought evaluation.

We conclude that a sub-catchment with the most frequent drought occurrence is generally associated with the longest duration, and consequently, the highest severity. We have noticed that sub-catchments experiencing the most frequent droughts do not always have the longest duration and vice versa; likewise, sub-catchments with the longest duration do not necessarily mean the highest severity and vice versa. All drought characteristics at different timescales must be integrated together for regional drought studies. Afghanistan, like many other semi-arid and arid regions, is prone to the impacts of a variable climate as water is a scarce resource in such regions due to the limited amounts of rainfall received annually and excessive evaporation resulting from high temperatures.

The strength of our current research which makes it outstanding of the previous studies in Afghanistan and surrounding regions is the sub-catchment scale understanding of the spatio-temporal dynamics of droughts. The use of sub-catchments as the spatial unit has improved the way that drought can be characterised spatially. It allows its input data and results to be spatially matched with and inform any catchment modelling activities undertaken in the country. The approach is also meaningful for both agricultural drought and hydrological drought and may provide insights into how the different types of droughts interact. Therefore, the results are crucial and practical to catchment planning for reliable adaptation and mitigation to drought impacts under changing climate conditions.

## Supporting information

**S1 File.**
(7Z)

**S2 File.**
(7Z)

**S3 File.**
(7Z)

## Acknowledgments

This study draws on data acquired through an Australia-Afghanistan research collaboration prior to the change of government in the Islamic Republic of Afghanistan in August 2021. The authors would also acknowledge and thank the Japanese government who, through its HYMEP projects, assisted the erstwhile Afghan government to record and quality check the climate and hydrology data stored in the AQUARIUS database.

## Author Contributions

**Conceptualization:** Yun Chen.

**Data curation:** Yun Chen, David Penton, Fazlul Karim, Peter Taylor.

**Formal analysis:** Yun Chen.

**Funding acquisition:** Shahriar Wahid.

**Investigation:** Yun Chen, Santosh Aryal.

**Methodology:** Yun Chen, David Penton, Fazlul Karim, Santosh Aryal.

**Project administration:** Shahriar Wahid, Susan M. Cuddy.

**Resources:** Shahriar Wahid.

**Software:** Yun Chen.

**Validation:** Yun Chen, David Penton.

**Visualization:** Yun Chen, Fazlul Karim.

**Writing – original draft:** Yun Chen.

**Writing – review & editing:** Yun Chen, David Penton, Shahriar Wahid, Peter Taylor, Susan M. Cuddy.

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
