## [Decision Letter · Decision Letter 0]

13 Sep 2022

PONE-D-22-16261Characterisation of meteorological drought at sub-catchment scale in Afghanistan using station-observed climate dataPLOS ONE

Dear Dr. Chen,

Thank you for submitting your manuscript to PLOS ONE. After careful consideration, we feel that it has merit but does not fully meet PLOS ONE’s publication criteria as it currently stands. Therefore, we invite you to submit a revised version of the manuscript that addresses the points raised during the review process.

We look forward to receiving your revised manuscript.

Kind regards,

Weili Duan, Ph.D

Academic Editor

PLOS ONE

Journal Requirements:

“No”

“This study was funded by the Australian Government and CSIRO and draws on data acquired a Australia-Afghanistan research collaboration prior to the change of government in the Islamic Republic of Afghanistan in August 2021. The authors do not have any communication or collaboration with the Taliban led government in Afghanistan and acknowledge and thank the Japanese government who, through its HYMEP projects, assisted the erstwhile Afghan government to record and quality check the climate and hydrology data stored in the AQUARIUS database.”

“No”

“No”

6. We note that Figures 1, 2, 3, 5, 6, 7, 8, 10 and 11 in your submission contain [map/satellite] images which may be copyrighted. All PLOS content is published under the Creative Commons Attribution License (CC BY 4.0), which means that the manuscript, images, and Supporting Information files will be freely available online, and any third party is permitted to access, download, copy, distribute, and use these materials in any way, even commercially, with proper attribution. For these reasons, we cannot publish previously copyrighted maps or satellite images created using proprietary data, such as Google software (Google Maps, Street View, and Earth). For more information, see our copyright guidelines: http://journals.plos.org/plosone/s/licenses-and-copyright.

    1. You may seek permission from the original copyright holder of Figures 1, 2, 3, 5, 6, 7, 8, 10 and 11 to publish the content specifically under the CC BY 4.0 license.  

Additional Editor Comments:

Please carefully improve the manuscript according to the all comments.

Reviewers' comments:

Reviewer's Responses to Questions

**Comments to the Author**

1. Is the manuscript technically sound, and do the data support the conclusions?

Reviewer #1: Yes

Reviewer #2: Yes

2. Has the statistical analysis been performed appropriately and rigorously? 

Reviewer #1: Yes

Reviewer #2: Yes

3. Have the authors made all data underlying the findings in their manuscript fully available?

Reviewer #1: Yes

Reviewer #2: No

4. Is the manuscript presented in an intelligible fashion and written in standard English?

Reviewer #1: Yes

Reviewer #2: Yes

5. Review Comments to the Author

Reviewer #1: The manuscript entitled " Characterization of meteorological drought at sub-catchment scale in Afghanistan using station-observed climate data” assessed the spatiotemporal of drought characteristic (distribution, frequency, severity and duration) over different sub-catchment of Afghanistan using Standardized Precipitation Evapotranspiration Index (SPEI) at different time scales. The paper is written well and investigated an important topic related to Afghanistan where such studies are very rare.

Reviewer #2: Comments to Authors

The manuscript “Characterisation of meteorological drought at sub-catchment scale in Afghanistan using station-observed climate data” presents an interesting research which is of importance to the area of study. I have the following comments and suggestions for the authors.

I suggest the authors add some numerical results such as some significant SPEI values obtained in the study area.

Introduction

Paragraph 1: Authors should highlight other drought occurences and their impacts in Afghanistan.

Paragraph 4: remove dimensional from the sentence.

In the second to the last paragraph, the authors discussed the constraints faced by the previous studies to include use of use of out-of-date long-term historical data; use of gridded global datasets at 0.5o (or ~ 50km) resolutions. Can the authors justify how these are constraints especially since some studies have used data of such resolutions and others have found gridded data sets to be able to replicate observed data sets. The following studies are examples

Changing characteristics of meteorological droughts in Nigeria during 1901–2010, https://doi.org/10.1016/j.atmosres.2019.03.010

Comparison of precipitation projections of CMIP5 and CMIP6 global climate models over Yulin, China, https://doi.org/10.1007/s00704-021-03823-6

Study area and data

Description of study area

Remove ‘at’ from “Afghanistan is located at in southern-central Asia (Figure 1).”

Figure 1 shows there are no existing gauge stations in most of the south of the study area, which may result into some catchments not having any stations within them. How were the authors able to capture the precipitation characteristics of the areas especially since the authors consider interpolation and use of gridded data sets as limitations in the introductory section?

I suggest some part of section 2.2.3 like the Hargreaves and Samani method especially the equations described be moved to the methodology section as they are more of method than data.

Also, why did the authors choose the Hargreaves and Samani method of estimating the PET since the Penman-Monthieth method is always recommended for SPEI calculation and since there can be significant differences amongst the SPEI series estimation by the different PET methods in some regions; like larger in semi-arid to mesic regions and smaller in humid regions.

Discussion

How has the authors tried to reduce the possible uncertainties reported in this section?

It would be interesting if the authors could add a paragraph or two discussing our the droughts in the country has affected agricultural output in the region.

6. PLOS authors have the option to publish the peer review history of their article (what does this mean?). If published, this will include your full peer review and any attached files.

Reviewer #1: No

Reviewer #2: No

---

## [Author Response · Author response to Decision Letter 0]

6 Oct 2022

Please refer to 'Response to Reviewers.docx'.

---

## [Editor Report · Decision Letter 1]

2 Jan 2023

Characterisation of meteorological drought at sub-catchment scale in Afghanistan using station-observed climate data

PONE-D-22-16261R1

Dear Dr. Yun Chen,

We’re pleased to inform you that your manuscript has been judged scientifically suitable for publication and will be formally accepted for publication once it meets all outstanding technical requirements.

Kind regards,

Weili Duan, Ph.D

Academic Editor

PLOS ONE

Additional Editor Comments (optional):

The original reviewer rejected or did not accept the review. I checked the response letter and the revision, and the existing problems have been dealt with very well. Personally, it could be accepted.
---

## [Editor Report · Acceptance letter]

27 Jan 2023

PONE-D-22-16261R1 

Characterisation of meteorological drought at sub-catchment scale in Afghanistan using station-observed climate data 

Dear Dr. Chen:

I'm pleased to inform you that your manuscript has been deemed suitable for publication in PLOS ONE. Congratulations! Your manuscript is now with our production department. 

Kind regards, 

on behalf of

Dr. Weili Duan 

Academic Editor

PLOS ONE